# Strategy in the Public and Private Sectors: Similarities, Differences and Changes

**John Alford** [1,*] **and Carsten Greve** [2]

[1]   The Australia and New Zealand School of Government PO Box 230 Carlton South VIC 3053, Australia
[2]   Department of Organization, Copenhagen Business School, Frederiksberg DK-2000, Denmark;
      cagr@ioa.cbs.dk
*   Correspondence: j.alford@anzsog.edu.au

**Abstract:** Strategic concepts and practices first evolved in the private sector, so they evoked much controversy when they migrated to the public sector from the late 1970s onwards. Partly this was about their (in)applicability to the distinctive features of government organizations, in particular their focus on public as well as private value, their situation in a political rather than a market environment, their almost exclusive capacity to use legal authority to achieve purposes, and the extent to which they often need to share power over personnel and resources with other public sector agencies. These and other factors complicated efforts to apply New Public Management and similar frameworks in strategy concepts in a governmental context. Partly also the traditional private-sector focus on single organizations did not resonate with the growth of network governance from the 1990s. The authors argue for an alternative model based primarily on the public value framework as a means of incorporating and going beyond traditional strategy thinking.

**Keywords:** strategy; public sector strategy; public value; authorizing environment; network governance; comparing public and private sectors

## 1. Introduction

"Given that most of the tools and concepts of strategic management were developed in the private sector, is it valid to apply them to the public sector?" That was the opening question in a chapter on strategy 15 years ago (Alford 2001). Even at that time, it was already a question that had been asked for many years before. Since then much has happened, so the nature of both the private and public sectors, and therefore the task of managing them, has changed greatly—but not necessarily in the same way across the sectors. In this article, we revisit the issue. We ask whether and to what extent private sector strategy concepts and techniques are suitable for the public sector, and in what ways these concepts and tools might be modified to take account of the public sector realities now. However, to answer these questions is to beg further questions.

For a start, there are many different strategic management tools and concepts—far too many to deal with adequately here (see (Alford 2001; Bryson 2004; Bovaird and Loffler 2003; Joyce 1999)). However, there is one framework that we argue is hegemonic in the field, which can legitimately be seen as representing the underlying logic of much of the strategy literature: the Harvard Business School's *Business Policy Model* (BPM) (Christensen et al. 1982). BPM is demonstrably the most widely read and enduring strategic management framework. In the 1980s, it gave birth to a putatively public-sector version, known as the *New Public Management* (NPM) (Hood 1991). By virtue of this heritage, we will argue, NPM can rightly be cast as a proxy for the use of private sector approaches in government.

BPM/NPM encompasses three factors that need to be considered in strategic management: what products to offer to what markets (the value to be created; the environment; and organizational

capabilities). These are explained in more detail below, but for the present, suffice to say that they constitute categories each with varying possible contents but comparable across the sectors.

At the same time, there is an emerging alternative model: the *Public Value* (PV) framework that starts from those categories, but provides a broader account of what sits in each category (Moore 1995, 2013). It therefore offers what might be called an "immanent critique" (Antonio 1981) of the dominant framework. Developed by Mark Moore and his colleagues in the Harvard Kennedy School[1], this framework both draws upon but also provides the most compelling critique of the BPM, and hence of deploying private sector techniques in public sector management. Because it is built on comparable categories, as will be explained, it facilitates identification of key similarities and differences between the sectors.

Thus, leaving aside its longstanding military antecedents, our attention here is to the nature of strategy in the BPM and the NPM, and to the public value framework. One important dimension is the extent to which the strategy model is oriented to content or to process (Alford 2001). The content role is about deciding what to do, utilizing concepts, analytical tools and organizational techniques (Andrews et al. 2006). It includes ideas of strategy like: setting long-term direction; and positioning, fit or alignment between purposes, means and the environment. This has been relatively significant in the private sector, and given rise to further content-focused strategic ideas (Boyne and Walker 2004; Andrews et al. 2009). The process role (Alford 2001) is one in which the issue is not so much what decisions might emanate from the strategic approach, as it is the pattern of deliberation. In the latter case, the task of the manager is less to find substantive solutions as such, but more to engage relevant actors to identify and deliberate about solutions and implementation opportunities. The key issues are such matters as: Who will take part in consultation or deliberations? Who will guide the proceedings? What information will be available? How much opportunity will each participant get to speak? Will proceedings be conducted in a large plenary or small groups or some other form? And many others . . .

Much of the research on strategy in the public sector tends to focus on the on the process role. A smaller proportion pays attention to the content role (e.g., (Vining 2016)), but tend to frame it in economic rather than broader terms. However, over recent decades, the content role became more important (from a lower base), but it augmented rather than displaced the process role, with the rise of public management interest in democratic deliberation, public participation, consultation, information-sharing and similar artifacts of democratic polities. Increasingly, public managers find themselves as initiators or responders, organizers, shapers, information-providers, advocates or devil's advocates in public deliberation processes. This raises normative questions as to the roles of public managers in democratic decision-making (Alford et al. 2016), but more important for our purposes are its ramifications for strategy and strategy-making. Is attention to the strategy process more necessary in the public sector? How much should public managers play a role in the political domain?

This article seeks to complement others in this special issue of *Administrative Sciences*. Whereas Andrews et al. (2017) elucidates the forms of strategy implementation in the public sector, we are concerned here with strategy-formulation. In addition, whereas Vining (2016) provides a useful elaboration of how various analytic constructs might apply to aspects of strategy in government organizations, we seek to compare how the various concepts might be put together as whole strategic frameworks, going beyond traditional public administration, New Public Management or collaborative network governance—with public value as an anchor for this reconsideration.

Thus, we analyze the antecedents and evolution of strategy-making, and apply them to two illustrative case studies. Using these categories, we take each of the elements of NPM, and then of public value management, and assess how appropriate they are to the public sector. We conclude that

---

[1]　　This notion of public value (singular) is not the same as the "public values" (plural) notion, which regards value as something akin to personally held norms and beliefs rather than "worth" or "utility"; see (Bozeman 2007).

the broad categories within the BPM are applicable to the distinctive characteristics of the public sector, but the contents of those categories mostly are not. For example, both have to understand and deal with their environment, but their environments are made up of different types of actors—the private sector facing a market environment while the public sector deals with a more political environment. By contrast to BPM, the public value framework is more attuned to the distinctive realities of the government sector. It therefore opens up a greater scope for action, and a framework for thinking about it.

However, the public sector is neither static nor monolithic. It is constantly changing, pushed by its tasks, its environment and the capacities it needs. The problems it faces in its social and natural environment—such as climate change, drug addiction, homelessness, or the need for economic restructuring—keep changing. At times their complexity and multi-faceted nature usually mean no single organization has the requisite knowledge, political standing or cultural insight to tackle them on its own. We therefore posit a third stage in public administration, after what we might call traditional public administration and corporate management: collaborative governance.

However, the private sector literature is not well suited to dealing with these kinds of changes. Its focus was on strategy for single organizations. This was certainly true when NPM was first introduced at the end of the 1970s; virtually all corporate strategizing started from the assumption that the organization had a unified purpose, and consequently often in competition with other single organizations. The purpose of strategy had been to gain a competitive advantage over other companies, not to build co-operative relationships with them (Brandenburger and Nalebuff 1996). However, as the 1990s began, it had already become abundantly clear that the model had to change to enable more multi-actor collaboration (Crosby and Bryson 2005).

In short, after first establishing NPM as a proxy for "private sector concepts and tools" in the next section, the section after that compares it with the alternative model, Public Value management. It then outlines the stage following NPM—multi-actor collaboration—and considers the extent to which it fits with each of the two frameworks.

## 2. The Rise of the Business Policy Model and New Public Management

Corporate strategy and its BPM version first took root in the United States in the 1960s, then spread to Britain, New Zealand, Australia and northern Europe, and from there has been taken up globally. On this point we argue that: (1) the BPM is and has been the dominant strand of corporate strategy thinking; (2) NPM is derivative of BPM; and (3) by corollary, applying NPM to government organizations is tantamount to installing private sector techniques and concepts into the public sector.

Strategy in either sector started from the commonsense notion of organizational planning, and indeed the very idea that managers might look ahead in deciding what to do—what Lindblom (1959) termed the "rational-comprehensive" approach. Perhaps because of its logical simplicity, rational planning has been enormously influential (Mintzberg 1988). However, it has also attracted criticism from various standpoints (starting with Lindblom's "science of mudding through"), notably for its insufficient recognition of the turbulence and complexity of the worlds in which organizations operate (Stoner et al. 1985; Robbins and Barnwell 2006).

A particular problem of the rational-comprehensive approach was its tendency to take the goal or purpose as given. There was not much structured thinking about what products the company should be offering to what markets; rather any change in the definition of the business tended to be incremental (Quinn 1980).

In response to these and other difficulties in recent decades, the strategy literature has proliferated. The sheer volume of publications in the field, and the contending schools of thought they articulate, makes it challenging to distil a uniform notion of strategy (Bryson 2011; Ferlie and Ongaro 2015; Fredriksson and Pallas 2016). However, within this literature BPM has tended to be hegemonic (Andrews 1971) in three respects. First, it is easily the most dominant intellectually. It shaped the common understandings of strategy analysis for decades from the start of the 1960s. According to

Mintzberg (1988), *Business Policy* (Christensen et al. 1982) is "one of the most influential" books in the field of strategy, with considerable impact on theorists and practitioners alike (De Wit and Meyer 2010; Macintosh and Maclean 2014). Many of the "alternative" theories and models in strategic management typically turn out to be elaborations of BPM—albeit useful in various ways. Second, it has also had a powerful and widespread effect on practitioners. It has been studied by hundreds of thousands of MBA students in North America and around the world, and some of its elaborated tools, such as SWOT analysis, are part of the language of business. Third, it is made up of elements—products/markets, environment, organizational capabilities—which can be framed both in generic and sector-specific terms that enable comparison. Here we use this model as the reference point, while also briefly canvassing key alternative schools of thought where relevant.

At the core of NPM was the notion of installing a strategic intent in public sector organizations, requiring and to some extent pushing them to articulate their purposes and drive, enable, empower and even inspire their departmental staff to pursue them. In short, NPM sought to encourage leadership as a prime responsibility for any agency manager. It also encouraged a variety of corporate planning features as well as a systematic management-by-results, or performance-based management, regime in the public sector. The emphasis was on "mission-driven" government, as Osborne and Gaebler (1992) put it in their well-known book *Reinventing Government*.

In particular, the focus was on positioning: the definition of what business a company is in, specifically, what products it offers to which markets. This definition is based on a judgment as to how well particular products and markets align with the market environment and organizational capabilities (Figure 1). Do these offerings constitute the most profitable value-proposition? Does the market environment show a substantial long-run growth in demand for those particular goods and services? Does the company uniquely or at least more advantageously possess particular capabilities that enable it to produce with enhanced value? In other words, do these three aspects—business-definition, market environment and organizational capabilities—fit together?

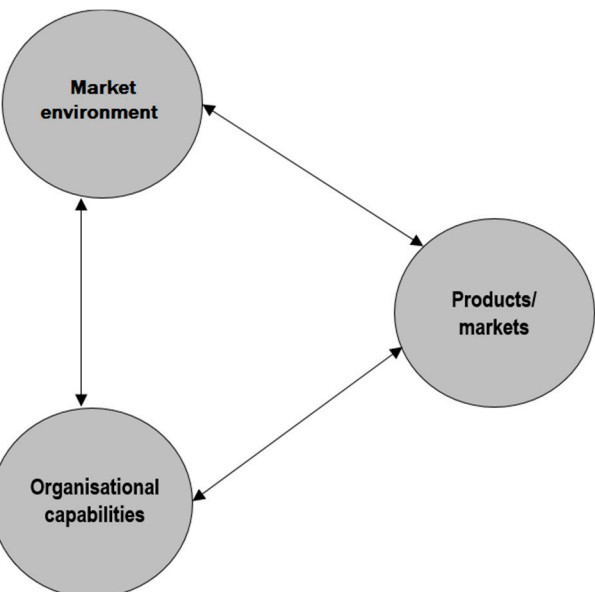

**Figure 1.** Strategic factors in the private sector.

Corporate management (in NPM form) was taken up by the public sector through various avenues from the late 1970s and early 1980s (Boston et al. 1996). One was via leading consulting firms who had worked in the Fortune 500 and imbibed their characteristic multi-divisional, performance driven corporate model, which they embraced as a "one best way" that they recommended to all their consulting clients (Saint-Martin 2004; Alford and Hughes 2008). Another was a desire to reap

efficiencies at a time when government was perceived as "too big", bolstered by an increasingly widespread belief that the private sector is inherently more efficient. The public sector's uptake of new management ideas continued in the 2000s and beyond with a focus on value creation, digitalization and involvement (Greve 2015).

The dominance of the BPM framework is further attested by the fact that most of the subsequent strategy models have turned out to be elaborations of BPM. An early idea about "defining the business" was Ansoff (1965) diversification matrix, by which strategists could analyze whether to deepen penetration in existing product-markets, expand or contract the business to produce new products for current markets or current products for new markets, or both (Abell and Hammond 1979). Tools for analyzing the environment came from Porter's focus on competitive advantage, from which the relative attractiveness of a given industry could be discerned (Porter 1980, 1985), and the Boston Consulting Group's "growth-share matrix", encompassing the growth of a particular industry and the market share of the company being analyzed (Abell and Hammond 1979). Finally, Harvard's Prahalad and Hamel (1990) considered the organizational capabilities as distinctive or core competences, as well as the latitude for encouraging "stretch" in those capabilities. Latterly, this category has included "dynamic capabilities", which Eisenhardt and Martin describe as

> 'the organisational and strategic routines by which firms achieve new resource configurations as markets emerge, collide, split, evolve and die' (Eisenhardt and Martin 2000, p. 1107).

It is not surprising, therefore, that a comparison of NPM's key elements with aspects of corporate management show a strong affinity between them, as Table 1 illustrates. The essential point is that all these concepts and techniques were derivative of the BPM, underscoring its centrality as a framework.

**Table 1.** New Public Management compared with Business Policy Model.

| Aspects of NPM | Fit with BPM |
|---|---|
| Hands-on professional management | Managers are trained in the theory and practice of management. |
| Stress on private-sector style of management practice | (Already incorporated into other elements of this table—overlaps substantially with hands on professional management). |
| Greater emphasis on output controls | Most Fortune 500 companies have multi-divisional structures with headquarters controlling divisions through output controls and performance measures. |
| Shift to disaggregation of units in the public sector | |
| Explicit standards and measures of performance | Performance measurement is embedded in the practice of management |
| Stress on greater discipline and parsimony in resource use | True of private sector firms pursuing cost leadership strategies (cf. Porter). Public sector finances determined more at budget level. |

*Source*: Part adapted from (Hood 1991). Also draws on (Porter 1985; Christensen et al. 1982).

However, a slightly tangential perspective—public choice—has become more salient as the reform impetus has progressed, even though that perspective has not figured much in research on strategy. Public choice can be seen as an extension of the logic of BPM/NPM. The latter entails separation of principals and agents (or purchasers and providers) *within* the organization (e.g., executive agencies), whereas the former involves widening the separation so the agent or provider is in a different organization from the principal or purchaser.

The NPM movement has witnessed a choir of critics over the years. Most are concerned with emphasizing that public sector bureaucrats are not managers or leaders as in the private sector, but civil servants in a rule-based system. Much of this traditional public administration critique concentrates on the differences between the two sectors. However, also important was that the business strategy model focused on the single organization. It did not take account of the fact that much corporate business entailed joint activity among a plurality of private, public or non-profit organizations—often referred to as network governance. This was becoming a burgeoning phenomenon as the 1990s wore on. Others

have feared that public trust is being undermined. In addition, recently, original NPM-researchers, including Christopher Hood, have explained how it got to cost more but did not work well. Their conclusion seems to be that NPM did not live up to its promises (Hood and Dixon 2015).

Thus, a good strategic approach is one that deploys what the private corporate sector has found to be useful—for example in articulating strategic intent, looking outward to the environment, or understanding incentives—while being cognizant of the distinctive features of the public sector.

## 3. The Public Value Framework

We argue that the public value framework is considerably better suited to these distinctive realities than any other model of strategic management, especially in terms of strategic content. First foreshadowed by Allison (1986) (see also (Heymann 1987)), developed and conceived largely by Moore (1995, 2013) it is now being taken up by increasing numbers of practitioners and scholars (O'Flynn 2007; Alford and O'Flynn 2009; Benington and Moore 2011; Bryson et al. 2014, 2015; Talbot 2009; Alford et al. 2017; Mulgan 2008; Barzelay and Campbell 2003; Llewellyn and Tappin 2003; Poister 2010; Meier et al. 2007; Weinberg and Leeman 2013). This framework acknowledges both the public and "business" imperatives that government organizations face. Its categories recognize that there are key issues affecting most if not all public sector organizations—such as what value should be produced, which aspects of our environment should give the most weight to, or whether we should deploy our own staff or externally hired ones. However, analysis of the specific content in each category shows up ways the public sector is different in important respects from the private.

What is telling about the framework is that it resonates both with practicing public sector managers and with scholars (although it is not without its critics—see (Dahl and Soss 2014)). Mark Moore (Moore 1995) fashioned it on the basis not only of his academic research but also of his teaching on executive education programs for public managers at Harvard's Kennedy School. Since then, the model has been taken up in numerous institutions and programs numbering in the tens of thousands or more. Moore's conception saw public managers as explorers who were constantly looking for new ways to create public value. He encapsulated the framework in his "strategic triangle", consisting of the value to be produced, the authorizing environment and the productive capabilities (see Figure 2).

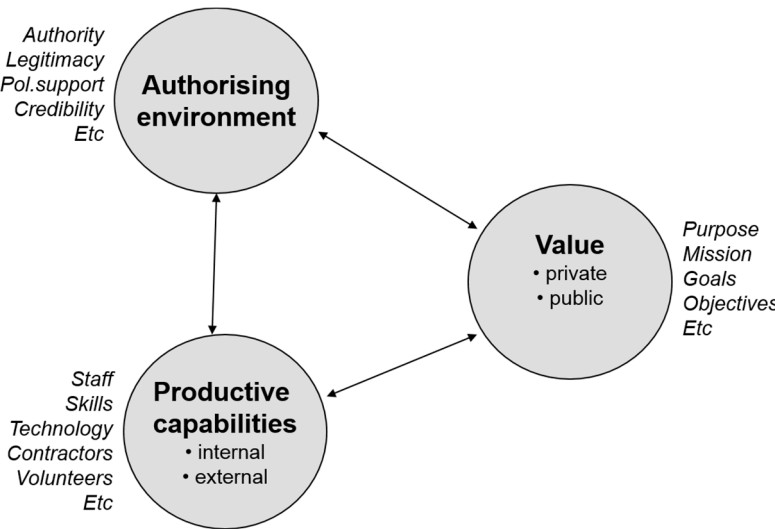

**Figure 2.** Strategic factors in the public sector.

First, the public manager is responsible for ensuring the production not only of private value but also public value—that which the citizenry receives collectively rather than consumes individually (for a more detailed discussion, see (Moore 2013)). Second, the environment surrounding a government organization is not a market environment but an authorizing environment, made up of various actors

who between them can provide the permission, resources and to some extent the capabilities the organization needs to conduct its work (Salamon 2002). Third, the means by which government organizations produce include various kinds of external providers, ranging from contractors providing specified services through co-producers and volunteers to external collaborations operating on the basis of trust and commitment (Alford and O'Flynn 2012).

The differences between the two sectors pose challenges not only for the applicability of corporate management to the public sector, but also to that of its offspring in that sector: the New Public Management. To the extent that it articulates a more elaborate conception of value, of the environment and of capabilities, public value thus constitutes a critique of NPM. However, while PV's conception of public management has been broader than NPM, it has tended to emulate NPM's focus on single organizations, perhaps because the stand-alone entity is its characteristic reference-point.

## 4. Comparing the Models to Public Sector Realities

The over-arching question, therefore, is how adequately this business framework characterizes public sector realities. Here we address the first of the two critiques of the BP model and of its cousin, NPM: its applicability to the requirements of the public sector (see Table 2). Such a comparison shows a modest degree of applicability of the Harvard model, mainly in form rather than in content. Each of the three elements can be seen as containing both private and public sector analogues; but they are different in substance. To explain this, we address two illustrative case studies.

**Table 2.** NPM compared with public sector.

| NPM Doctrinal Component | (Lack of) Fit with Public Sector |
| --- | --- |
| Hands-on professional management | Yes, if managers trained in public sector management practices like their private sector counterparts. |
| Explicit standards and measures of performance | Difficult (if not impossible) in many parts of public sector—problems of uncertainty, information asymmetry, interdependence, asset specificity. |
| Greater emphasis on output controls | |
| Shift to disaggregation of units in the public sector | Interdependence of public functions can make disaggregation and aggregation complex. |
| Stress on private-sector style of management practice | Managers' attitudes/culture may be out of sync with public purposes. |
| Stress on greater discipline and parsimony in resource use | Finances determined more at whole-of-govt level and imposed on divisions. |

### 4.1. Case 1: The New Zealand Department of Corrections

In November 2008, New Zealand's Department of Corrections (DoC) faced a dilemma[2]. A new coalition government led by the center-right National Party had just been elected and was seeking a dramatic reorientation of the justice system. There had been a 50% growth over the last decade in the number of prisoners, as well as those serving community-based sentences, stretching the prison system to breaking point. Current prison capacity was projected to be fully utilized by 2010, with serious ramifications for offenders, staff and the community. However, the new government's "tough-on-crime" policies were likely to aggravate this problem. It had pledged to tighten bail laws, abolish parole for violent repeat offenders, toughen sentences for child abuse and gang-related offences, and review home detention in sex, drugs and violence cases. At the same time, the new government had pledged a crackdown on state services spending, stating that it would not consider any "budget bids" for new funding. An economy in recession and lingering uncertainty after the 2008 global economic crisis led Treasury to recommend public sector restraint as well.

---

[2]    This account of the Department of Corrections case is from the ANZSOG Case Library (Padula 2013).

### 4.2. Case 2: Danish Decommissioning

The official mandate of Danish Decommissioning (DD) is to dismantle and clean up after a nuclear test facility in Denmark and to contribute to a medium- and long-term solution for Danish radioactive waste. This follows a decision in Parliament to abandon nuclear power in Denmark. The focal public manager here is the management of Danish Decommissioning. Although the decision is taken at a senior bureaucratic and political level, there is no concrete resolution yet as to where in Denmark to put the radioactive waste.

The value to be produced is the final outcome of storing the radioactive waste material in a safe manner. The authorizing environment is relatively complex, with different individuals, groups and organizations having a stake in the final decision—indeed, a significant and long-term stake. At the apex of the formal authorizing environment are the Danish Parliament and the Danish Government. However, equally important is the informal authorizing environment: to do its job Danish Decommissioning has to collaborate or at least interact with a wide range of organizations, both international organizations and Danish local governments of which some will eventually be chosen for the "medium-term" store location (100 years) or "long-term" store location (indefinitely). In this collaboration, the various parties surrounding the organization are not only political stakeholders in the sense of helping to provide legitimacy and support to agency and purposes, but also in the same process contribute co-productive effort to help achieve those purposes. Central to DD achieving its mandated purpose is to engage with the various stakeholders, navigating between complaining residents and undecided politicians in Parliament and in Government, elucidating an approach that attracts sufficient consensus to be feasible. In effect, DD has to collaborate with external actors not only in implementation of the task, but also in developing the strategy for doing so. DD cannot simply hand down a strategy from on high and insist that everyone follow it; it has to ensure that all the relevant parties see it as their own—a task made all the more fraught by the potentially massive hazards of nuclear power.

## 5. Applying the Concepts: An Analysis of Two Cases

How applicable to DoC and DD are the two frameworks—BPM and NPM—described above? The short answer is: only to a limited extent. First, both DoC's and DD's services sit awkwardly with this private sector idea, at both the strategic and the transactional levels. DoC's aim is to contribute to the overall Justice Sector outcome of a "safe and just society" through "upholding the integrity of sentences and orders", "reducing re-offending", and managing offenders "safely and humanely". However, it does not have customers in the same sense as private companies do. The recipient of the services does not pay any money for them. At the same time, they typically do not "enjoy" the services. Indeed, they are disadvantaged for the sake of the broader citizenry, which jails convicted offenders either to contain them, to deter them, to rehabilitate them, or in some cases simply to enact retribution. In the DD case, a nuclear cleanup is a profoundly collective good affecting everyone within a radius (and beyond), even if they do not want it to.

All of this means that at the strategic level, defining the business is both complex and contentious. It is complex because it can be difficult to discern causal connections between applying particular punishments, rewards or training opportunities to prisoners and their subsequent behaviors. It is contentious because there are competing schools of thought about what is most effective in dealing with offenders. In New Zealand this debate was a live issue, in which politically conservative but populist media on "talk-back" radio ("shock jocks") voiced extreme views. Some scholars described their approach as "penal populism" (Pratt 2007).

Further complicating this is that much of the value that emanates from the operation of both DoC and DD is collective in nature. Part of this is instrumental: it provides remedies to market failure, such as "the provision of a public prison system is good upholding the integrity of sentences and orders" and "reducing re-offending". Part of it was deontological (Moore 2013), in managing offenders "safely and humanely". These are all instances of public value (Moore 1995, 2013), which is "consumed" by

the collective citizenry as a whole, and refers not only to the outcomes of value-creation or production processes, but also to the institutional channels through which they are discerned. DD is obligated by law to grapple with the task of nuclear waste and the responsibility cannot be passed on to an organization on the market. One significant consequence of this is that the organization may not have the flexibility to withdraw from such products/markets.

In any case, it is clear that there are fundamental differences of opinion about the best approach to dealing with offenders, ranging from incarceration to rehabilitation, and to questions of where to put the nuclear waste. To ask which is likely to be more valuable begs the question of what "valuable" means in this context. For the sake of simplicity, we adopt the corrections department's own over-arching goal of "a safe and just society", which translates at an operational level to "crime reduced", which in turn requires reduction in various factors that influence crime rates (Department of Corrections 2008). The question, therefore, is whether crime is more effectively reduced by incarceration or rehabilitation. For the DD case the purpose is environmentally responsible decommissioning that is the official value provided.

The extensive research about dealing with offenders seems at first sight to support the argument backed by many studies, that increased incarceration reduces crime (for a comprehensive survey, see (Stemen 2007)). Since the society as a whole benefits from decreased crime, it follows that, all else being equal, more punitive treatments are valuable. However, the association is fairly weak: a 10% growth in incarceration generates only a 2–4% decrease in crime (Spelman 2000). Also, the studies vary widely in their estimates (Stemen 2007). This picture will get worse as the prison population burgeons. As one senior scholar in this field summarized a review of numerous studies: "analysts agree with apparent unanimity that future increases in incarceration rates for such offenders will do less and cost more" (Stemen 2007).

Thus, the research at large is inconclusive. There is, however, a more relevant answer embedded in the department itself, from looking at what was actually happening. In an election won by Labor in 1999, a citizen-initiated referendum had called for a "tough-on-crime" approach, which the government began to implement. However, the overall result of this incarceration focus was that the prison population increased by 50% between 1999 and 2008, and was forecast to increase a further 20% by 2016. Subsequent attempts to apply new community sentences also did not succeed.

It is, therefore, reasonable to argue that rehabilitation is of value to the public—not only was it effective, but also it was considerably cheaper: it cost over $90,000 per year to keep an offender in prison, whereas home detention cost $25,000, and a community work order $2000. On a spectrum from rehabilitation to incarceration, the most valuable strategy for the Corrections Department is towards the less punitive end.

However, if rehabilitation is so effective, why have governments traditionally shied away from it? The answer lies in the nature of the authorizing environment, the second factor, to which we now turn. The assumption that the organization is surrounded by a market environment needs revision. Instead, the prison system is situated in a political environment—what Moore (1995) calls an *authorizing environment*. This is similar to a stakeholder environment. It comprises all those actors who provide permission and/or resources to enable the organization to function, and thereby have a stake in its operation. Thus, the director-general of the Corrections Department is subject to an authorizing environment which includes, *inter alia*: the prime minister; the minister for corrections; NZ police; political parties; the Treasury; parliament; prison officers and their union; lobby groups; the judiciary; and to a certain extent the media. These stakeholders include not only those with formal authority over the Corrections Department—such as the minister—but also those with informal, small-"p" authority and influence. Analysis of this environment enables judgment about where the "balance of forces" lies in respect of their interests and power. An environment populated by a certain set of actors may

be more or less amenable to a particular position. This has important implications for the issue of alignment between the three elements, to be discussed below[3].

This relates to the third assumption of the BP model, concerning the organization's capabilities. Corrections had 7000 full-time equivalent (FTE) staff, over half of them in prison services and 30% Māori or Pacific people. Although it was true that the department had command managerial authority over these staff, they were insufficient and inexperienced. Consequently, Corrections also drew upon external parties such as volunteers, non-profits and regulatees. It had approached other government departments and community organizations to look at ways of bolstering re-integration efforts. To the extent that they did so voluntarily and with competence, these external providers contributed valuable inputs or "co-production", and, at a higher level, inter-organizational collaboration and co-operation. However, from the point of view of the organization, managing them calls for the exercise of influence, persuasion, negotiation and leadership rather than command.

The application of the PV model has broadened over time in its view of organizational capabilities and stretching it towards a more network governance approach.

For the DD case, the government organization had expertise and qualified personnel at its disposal, including experts on nuclear issues and long-term waste solutions, but DD also drew on expertise from other areas of government, from the private sector, and from the international cooperation bodies on dealing with nuclear waste.

At this point, we can consider the notion of alignment in a public sector context (see Figure 2). It is clear the three elements—the products/markets or value, the political environment, and the operational capabilities—do not line up very well with each other in the case of corrections. First, there is contention about the appropriate outcomes to seek—notably between the rehabilitation and punishment perspectives. However, regardless of the vehemence of populist tabloids, on balance the research evidence seems to lend weight to the rehabilitative approach. That is, if the objective is to reduce crime, the activities that rehabilitate and reintegrate offenders into mainstream society can be said to be more valuable, both for the public and individuals involved.

However, if we turn to the authorizing environment, we find a mismatch. The reason is that the dominant position in that environment overall is support for the punitive approach—and indeed of antipathy to the rehabilitative one. This occurs for a number of reasons, including: "penal populism"; racism, fear of crime (whether justified or not); and all of this sharpened and disseminated by the tabloid media. Whatever the reason, it leads to a profound misalignment between the value-proposition and the authorizing environment. The most valuable course of action for the department to take happens also to be the least politically acceptable. In this situation in the private sector, whether to invest in a particular product/market would be subject to an analysis of risk vs. return: the higher the risk, the greater the return expected, to compensate for the risk. These notions are not quite the same in the public sector, but we can frame an understanding through related concepts. Thus, "return" could be constructed as analogous to public value, while "risk" is political risk—the risk of failing to prosecute the policy debate and consequently of losing "clout" (or even one's position). Because of the previously mentioned difficulties in specifying and measuring in the public sector, determining appropriate objectives calls for well-honed judgment. However, this is no more challenging than the major policy decisions senior bureaucrats must make in alternative contexts.

The Department's organizational capabilities are also inadequate for the rehabilitative strategy. There are too few prison officers for the task, with demand expected to exceed supply soon; their ethnic profile does not match that of the prisoners, who have more Maori than do the prison officers; the physical accommodation is old and falling apart; and many of the prison officers were insufficiently experienced for the job.

---

[3]  It is true that some corrections agencies, not only in NZ but also in Australia, the UK and the USA, have contracted out some of their prison services through competitive tendering, but this hasn't altered the basic function of the prisons in incarcerating those who most approximate private sector customers, and the status of those imprisoned.

Thus, the strategy that would be most valuable for the organization to pursue—rehabilitation—struggles to line up with both the political environment and the available capabilities. Without such alignment, even if only partially, the strategy will lack the permission, resources and capabilities needed to put it into play.

Stated this way, it may seem like an impossible choice: either pursue the less valuable approach, leaving substantial value uncreated—even if urgently needed, such as refurbishing some of the prisons—or prosecute the evidence-based case for what is more valuable, and come into conflict with powerful forces leading potentially to the loss of one's position. However, in fact the very things about the public sector that make it apparently problematic to apply private sector concepts can also be recast as opportunities.

First, the indeterminacy of public value is a two-edged sword. It enables some public managers to avoid being pinned down with responsibility for outcomes, hiding behind vague measures. However, it can also enable managers to be more flexible in complex, rapidly changing situations. Second, the contested nature of the authorizing environment, to the extent that it entails debate and deliberation, can help improve the quality, responsiveness and realism of policy decisions. This is redolent of the notion of political capital, which can be built up, invested or spent analogously to financial capital. Third, the presence of third party actors in the production of value opens possibilities for extending and improving production, including when resources are short.

Finally, the notion of alignment can offer a way of thinking about precisely the kind of dilemma faced by the Corrections department. The key to this is to recognize that alignment can be calibrated—a value-proposition can be more or less aligned with its authorizing environment or its capabilities. This means that it is possible to delineate alternative options—specifically ones that partially meet what is required, but even better that they can be combined with other measures that enhance the impact of these alternatives. There are also various ways of discovering these opportunities—many of them derived from the public sector.

Typically, managers scope the options by framing those at each end of the possible range—for instance, a "do-nothing" option or the "do-everything" option. Neither is likely in itself to find favor with a majority of the authorizing environment. However, they are useful as heuristic devices.

However a more finely grained analysis reveals possibilities that may be viable as well as valuable. This requires unbundling the departmental "product" offerings as well as segmenting its markets, rather than planning on the basis of "one-size-fits-all" services. Specifically, understanding the compliance postures (Braithwaite et al. 1994) of the various offender segments allows better tailoring of services—and of penalties—to the factors motivating the behavior of each segment, making them more effective as well as cheaper. For example, a portfolio of responses based on offenders' past history might include:

- Being tough on violent or sex offenders, but providing counseling, support, training and other rehabilitative methods to prisoners assessed as having potential for reform.
- Streamlining the "front end" of the system to make the process of accepting an incoming offender quicker and more efficient.
- Addressing drug and alcohol dependency.
- Providing post-release services, especially by working with other agencies to deliver accommodation, employment and other resettlement services.

These responses would contribute to reduced recidivism, especially by Maori/Pacific Islanders. They would also lend strength to a long-term effort to tackle the causes of crime, in employment, education, dependencies, land, identity, etc.

Thus, the form of the BP model provides apposite categories within which to organize a strategic analysis—in either the private or the public sectors. However, the content considered in that model needs to be amended if it is to have relevance to the public sector. Without adjusting the model to take account of the distinctive realities of government organizations, it is an alien implant that not only

fails to comprehend the public sector, but may also distort the understandings of those wielding it in a manner that is positively harmful.

## 6. Pursuing Public Value in a Governance Perspective

However, the differences between public and private sector organizations have not been the only stumbling blocks to effectively implementing corporate strategy in government. Also emerging as an issue is another factor: that in many settings, multiple organizations have shared power with typically diverse issues and stakeholders, where no-one is wholly in charge (Crosby and Bryson 2005). These are situations where the single organization—no matter how coherent its own strategy—has only limited leverage unless it can persuade, maneuver, push, educate or financially reward the other actors to go along with it. These situations arise from or with the proliferation of increasingly complex problems such as climate change, homelessness or drug addiction. The intractability of some of these problems may mean there is no consensus about cause or remedy; they are sometimes referred to as "wicked problems" (Rittel and Webber 1973; Head and Alford 2014). The practical import of this is that a single leader of a single organization—even a bold, imaginative, influential one—finds it difficult to unearth causes, develop solutions and get others to implement them. In short, the essential unit of analysis is not the organization but the network, itself made up of plural organizations. Another contributory factor is growing popular enthusiasm for democratic participation, fuelled in part by greater access to education. Our second case study, Danish Decommissioning, sheds more light on the implications of multi-organizational circumstances for strategy.

The DD case shows how the management of the organization has expertise inside the government organization, but also seeks to cooperate and engage with stakeholders in the environment. DD has to seek knowledge and information from outside sources in national and international knowledge networks. DD also has to be prepared to engage with organizations and persons in all those areas in Denmark that will be affected by the decision of where to locate the nuclear waste facilities. Therefore, DD has been active in meetings around the country, and by soliciting opinions, views and not least evidence for what type of nuclear waste facility is most safe and secure.

This networked governance contribution to the literature was partly a reaction to the "contractualist" flavor of NPM, with its emphasis on tight specification and monitoring backed up by sanctions and rewards—a reaction which constituted an implicit assertion of the inherent differences between public and private sector management. However, it also filled a gap in the PV literature, albeit not an insurmountable one: that PV had mostly been applied so far only to single actors and organizations. However, recent years have seen not only a flowering of efforts to apply PV to networks, but also some exploration of how understanding networks might shed more light on PV, especially the significance of its process role (Geuijen et al. 2017).

None of this is to suggest that networks are problematic for PV. The lack of interaction so far of networks and PV with each other is not because they are contradictory, but primarily because there has been little effort to apply them to each other. It seems likely that PV and network governance are positively compatible with each other. Led by a variety of scholars (see Klijn and Koppenjan 2015 for an overview), the network theorists argued that the public sector consisted of aligned organizations in networks, not stand-alone public sector organizations which could optimize their own strategy in seclusion. The network scholars pointed to the opportunity for managing across organizations in networks in order to achieve publicly and politically defined goals. The strategic perspective shifted to what Klijn and Koppenjan (2000) referred to as management of network structures, and managing within networks, to align certain organizations towards a common goal. The new incarnation of the network approach can be found in scholars who conceptualize "collaborative governance" as the key term (O'Flynn and Wanna 2008; Donahue and Zeckhauser 2011).

The network- and collaborative-governance school has also had its fair share of criticisms. Among the most important are that networks become very complicated and that they entail huge transaction costs. Many of the most recent additions to the public sector literature appear to combine some of the

schools mentioned above, especially the public value creation school and the network and collaborative governance school. Scholars like Bryson, Crosby, Stone, Bloomberg (Bryson et al. 2015), Page et al. (2015), Geuijen et al. (2017) and others have formulated an argument about public value creation in cross-sector settings. Most recently the term "public value governance" has been aired. Whatever the term, the key implication appears to be that the strategy of public value creation that Moore and others talked about is seen from not just the individual public manager's or public organization's point of view, but is more something that organizations in networks are preoccupied with. Strategic intent and public value creation increasingly takes place in a networked or collaborative governance environment. Strategy is something that organizations formulate and implement together, and has become more explorative. The Danish Decommissioning case illustrates this argument.

## 7. Discussion and Conclusions

Being strategic in the public sector often entails the manager having to deal with politics while not being seen to step outside their constitutionally assigned roles. This calls for political astuteness of a high order (Hartley et al. 2015). Public managers should strive to align the different elements sketched out in Moore's PV framework, to get the relevant stakeholders on the same page in addressing important issues. As we discussed, alignment here is not so much about finding the "perfect" or "optimal" solutions, but rather to make clear what the trade-offs are in certain decisions and approaches. To calibrate possible alignment scenarios is one of the key activities that public managers can engage in. Public value creation must be seen in a collaborative governance perspective. Organizations cannot optimize strategic intent on their own, as some of the NPM-literature seemed to assume, but they must increasingly work with other stakeholders and also citizens in wider democratic processes to achieve forward momentum.

To be sure, there are approaches in the literature on public management and governance where a process perspective has been highlighted previously—think, for example, about "management by groping along" (Behn 1988), and the original incremental model of decision making proposed by Lindblom (1959), Simon (1947) and others. We do not feel that we are saying something completely different here, but rather that we re-focus the public sector strategy debate on some of the essential actions that emphasize step-by-step process-based aspects of decision-making and implementation.

The analysis of the strategy of New Zealand's Department of Correction suggested that the government department acted under different constraints than a private sector organization might have experienced, and that there were many responses that the Department could choose from, but ultimately it had to engage other stakeholders in them for legitimacy purposes. In correctional activities, alignment could be calibrated from less to more optimal ways to address the issue, but these alternative ways resonated differently with the department's authorizing environment. The department had to find ways to trade off the likely value to the public against the constraints of both its authorizing environment and productive capabilities. This could mean varying one or both of those factors. The ability to influence these factors varied: easiest to change was the value-proposition that the organization was pursuing, while most difficult was the authorizing environment, with productive capabilities somewhere in between.

Also very relevant here is the temporal dimension: solutions to dysfunction in one factor might aggravate another over time. How a decision affects a situation at one point in time might influence how it would be viewed in the future. For example, an expeditious "ride over the delays" approach might get the job done, but offend many of the actors involved in the long run. Public managers must try to balance the different viewpoints, but come up with adequate, possible responses for which others can take responsibility.

The analysis of the Danish Decommissioning case demonstrated the significance of creating a legitimate process. Addressing the issue of nuclear waste disposal takes a long time and involves several steps along the way. It means developing a proper hearing process for the local communities involved. It means designing a process for involving the local governments and the mayors. It means

designing a broader communication policy that spans the wider public, who may be interested in the subject as long as it means that their locality is avoided for choice of storage place. Designing and managing the process is the all-important management task. This also implies efforts in "managing upwards" in the sense that the political process itself with politicians involved has to be seen as fair and honoring various democratic principles. This can constitute a tricky task, since at times the line between the political and the administrative realms is very blurred. To navigate this, managers need to exercise judgment about how far to push the political dimension for the sake of creating substantive value for the public.

Strategy in the public sector is moving from being mostly about corporate planning models in single public sector organizations in the NPM-era to a broader and more widely dispersed activity that several managers and organizations participate in. A way forward for public managers is now to be less focused on the content of strategy in the sense of finding the right performance and optimized outcome. Instead public managers must pay more attention to forming and shaping public value-propositions, and to engage with other stakeholders and citizen participants in doing so. Politicians, stakeholders and citizens all play a part in realizing the goals of a strategy. Therefore, a more balanced route for public managers is to view themselves as partners in public value creation.

**Author Contributions:** The authors contributed equally to this paper.

**Conflicts of Interest:** The authors declare no conflicts of interest.

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
