# Peer review of "Strategy in the Public and Private Sectors: Similarities, Differences and Changes"

_admsci, doi:10.3390/admsci7040035_

Round 1

Reviewer 1 Report

Thanks for the revised manuscript. The paper addresses my earlier issues well and is suitable for publication. Interesting read and a valuable contribution to the debate about strategy in the public sector.

Author Response

Thank you for your encouraging response, as well as your suggestions in an earlier round  of reviews.

Reviewer 2 Report

The paper provides an insightful overview and review of the meaning of strategic management for public services organisations. It is well written and shows mastery of the topic. There are three concerns I would raise, hoping these considerations may be helpful for the further improvement of the paper. First, while the individual claims made throughout the paper are all in themselves agreeable and well discussed, the reader is left wondering what is ultimately new that is being added to the literature by this paper, All claims, while sensible, are hardly new and innovative. Second, the lineage between BPM and NPM - apart from being hardy new - is questionable, as the argument seems to emphasise the 'business-like government' component of NPM, but as a minimum the public choice component should also be considered. Third, the two cases seem more didactic/illustrative than research-intensive empirical studies; while it is fully understood word number constraints limit the thickness of the narrative that it is possible to develop in the length of a paper, the two cases seem to be taken for illustrative purposes more than for theory testing or theory enhancing (or theory falsification). Finally, the argument about the superiority of public value/public value governance perspective is more asserted than demonstrated, and here again it is hardly new, or hard to link to and underpin based on the findings of the two cases.

Author Response

Authors’ response to reviewers

Reviewer #2

We thank Reviewer #2 for the suggestions made, which are helpful to us in understanding what we need to clarify. The reviewer raises four points:

The first is to question whether the article adds anything new to the literature. Our view is that with the exception of Moore (1995) and perhaps Mulgan (The Art of Public Strategy, 2009), little has been written about the content of strategy in the public sector, as opposed to the process of strategy-making, and also as opposed to the strategic dimensions of the public value framework. The article surfaces some conceptual tools (pp. 6, 9-13) for ascertaining what strategic choices would be valuable in a given context.

Reviewer 2’s second point is that the lineage between BPM and NPM is questionable because it neglects public choice versions of strategy. Our view is that the BPM/NPM link is strong – and our position is supported by the discussion on pp. 3-6 – but that we agree with the reviewer that in addition there is also room to consider the public choice perspective, even though that perspective has not figured much in research on strategy. We see public choice as an extension of the logic of BPM/NPM. The latter entails a more decoupled kind of structure, in which principals and agents (or purchasers and providers) are more at arm’s length from each, whether within the organisation (e.g. executive agencies), or between the organisation and external actors. We have added this point to the discussion on p. 6.

Reviewer 2’s third point is that the two cases are didactic/illustrative, not theory testing or enhancing. In fact, that is what we intended: we argue that the cases are meant to be illustrative, to enable better understanding of a multi-faceted argument.

Finally, Reviewer 2 argues that the ‘superiority of public value/public governance perspective is more asserted than demonstrated’. We contend that this is not an empirical study, but rather a clarifying and laying out of a position. That position is supported by a growing body of argument from the literature.